# An Instrument to Measure Mental Health Professionals’ Beliefs and Attitudes towards Service Users’ Rights

**DOI:** 10.3390/ijerph16020244

**Published:** 2019-01-16

**Authors:** Francisco José Eiroa-Orosa, Laura Limiñana-Bravo

**Affiliations:** 1Section of Personality, Assessment and Psychological Treatment, Department of Clinical Psychology and Psychobiology, Faculty of Psychology, University of Barcelona, Barcelona, 08035 Catalonia, Spain; lauralb93@hotmail.com; 2Yale Program for Recovery and Community Health, Department of Psychiatry, Yale School of Medicine, Yale University, New Haven, CT 06513, USA; 3First-Person Research Group, Veus, Catalan Federation of 1st Person Mental Health Organisations, Barcelona, 08025 Catalonia, Spain

**Keywords:** mental health services, attitudes, beliefs, coercion, paternalism, discrimination

## Abstract

We aimed at developing and validating a scale on the beliefs and attitudes of mental health professionals towards services users’ rights in order to provide a valid evaluation instrument for training activities with heterogeneous mental health professional groups. Items were extracted from a review of previous instruments, as well as from several focus groups which have been conducted with different mental health stakeholders, including mental health service users. The preliminary scale consisted of 44 items and was administered to 480 mental health professionals. After eliminating non-discriminant and low weighting items, a final scale of 25 items was obtained. Exploratory and confirmatory factor analyses produced a four-factor solution consisting of the following four dimensions; *system criticism*/*justifying beliefs*, *freedom*/*coercion*, *empowerment*/*paternalism*, and *tolerance*/*discrimination*. The scale shows high concordance with our theoretical model as well as adequate parameters of explained variance, model fit, and internal reliability. Additional work is required to assess the cultural equivalence and psychometrics of this tool in other settings and populations, including health students.

## 1. Introduction

The mental health sector has undergone two fundamental transformations in the last half century, namely *Deinstitutionalization* and *Recovery* [1]. Both processes have involved an increase in service users’ autonomy and freedom of choice. The *Recovery* movement has mostly been driven by service users themselves, which has entailed a significant increase in participation, and consequently a reduction in paternalistic behaviours carried out by professionals. These processes have also improved the professionals’ awareness of service users rights, and have led to a reduction of coercive measures and a shift from symptom reduction to rehabilitative and recovery approaches [2].

Despite all these improvements, many service users still report stigmatizing attitudes, including professional paternalism and emotional estrangement [3,4]. Therefore, receiving a mental health diagnosis is still considered as a predisposing factor that can lead to the experience of stigma from both the social environment [5] and mental health professionals [6].

Stereotypes depicting mental health service users as incompetent, weak, incurable, and violent lead to social discrimination and coercive professional practices [7]. Some examples are involuntary inpatient and outpatient treatments, forced medication, overmedication, electroconvulsive therapy under duress, mechanical restraints, seclusion, isolation, and arbitrary legal incapacitations and guardianships. The underlying beliefs that influence the decisions that professionals make appear to be formed by a lack of awareness of one’s own prejudices [4]. In this sense, the perception by some professionals that the moral side of their decisions is not relevant to the recovery process may contribute to the acceptance of coercion as a standard practice [8].

Given the extent of the consequences of stigma and coercion towards mental health service users, it is essential to raise awareness among mental health professionals in order to foster non-stigmatizing and empowering attitudes through frameworks such as *Recovery* [9] and *Citizenship* [10]. In the context of planning and implementation of these training and awareness activities, there is a need to evaluate the impact this has on the beliefs and attitudes of professionals through standardised measures.

### Previous Measures

The first scale that included beliefs and attitudes among mental health professionals towards mental health service users was developed by Gilbert and Levinson [11]. This scale was used as a method to understand mental health practices on a continuum from custodial to humanistic [12]. Another scale dealing with professionals opinions about mental illness was validated shortly afterwards [13]. Alongside these early developments, Goffman re-defined the word “stigma” to refer to a non-physical, invisible signal, making a person’s social status undesirable [14]. Relatedly, different mental health stigma measures have been developed and are usually applied to the general public [15,16,17,18,19,20,21,22], mental health service users [23], and some others to professional audiences [24,25,26,27,28,29]. Another issue that has also generated assessment tools is the stigma perceived by those in the medical profession in general towards psychiatry in particular [30].

Over the last decades, in parallel to the rise of the *Rehabilitation* and *Recovery* movements, professionals have become more conscious of the need to offer a non-discriminant care based on users’ rights. In this context, measures on recovery-based knowledge [31,32,33], attitudes [34], expectations [35], and practices [36,37,38,39,40,41,42] have been developed (see Table 1). These instruments have enabled the evaluation of dozens of projects which have implemented the philosophy of recovery in many health institutions. However, there are also limitations of these instruments, including the impossibility of administering the attitudes and knowledge measures at baseline with lay professionals, given that they assume a certain degree of knowledge of the principles of recovery. Furthermore, all current professional stigma instruments are designed for certain professional groups [24,25,26,29] or mental health conditions [27,28]. Finally, scales measuring recovery practices assume that some level of implementation of these practices has already been done. 

So far, there is a lack of multidimensional measures that can be used to assess training and awareness activities attended by different mental health professional groups with heterogeneous levels of knowledge and awareness of the importance of respecting service users’ rights. For this reason, the objective of this work is the development of a flexible instrument in order to measure the beliefs and attitudes related to service users’ rights among all types of mental health professionals.

## 2. Materials and Methods 

### 2.1. Development

An initial set of 44 items was developed in Spanish by researchers with a lived experience of mental health problems. This set was reviewed by Catalan experts on stigma awareness and community mental health including board members of the Catalan Federation of First-Person Mental Health Organisations, where all the members have lived experience of mental health problems. Half of these items were derived from a systematic review of previous measures (see Table 1). We reviewed scales on stigmatizing attitudes and beliefs as well as recovery-based knowledge and practices. We found items related to three cognitive levels: attitudes, awareness and knowledge, and four thematic domains: empowerment/paternalism, recovery, stigma, and rights.

Additionally, we conducted 11 focus groups with mental health professionals, 7 focus groups with service users, and 1 focus group with relatives of service users. The first six groups carried with mental health professionals and those carried with service users and relatives were used for the adaptation to the current context of items based on the literature and the creation of a pool of 22 completely new items. In the last five groups carried with mental health professionals, the scale was presented at the beginning of the session, leaving time for participants to respond. During the discussion, professionals could comment on the content and contextualise how they had answered the items.

The order of the items was randomised before starting the administration of the scale. Regarding the anchor points of the scale, we chose a four-point Likert scale (I fully disagree, I disagree, I agree, I fully agree). This mode helps to minimise middle response bias, which is frequent in attitude research [43].

### 2.2. Sample and Procedure

The sample used for the psychometric validation of the Beliefs and Attitudes towards Mental Health Service Users’ Rights Scale (BAMHS) was comprised of a total of 480 Spanish-speaking mental health professionals. These professionals worked in a diverse range of settings, including inpatient care, outpatient care, rehabilitation, supported work, leisure and free time services, etc. Among these, there were psychologists (29%), mental health nurses (15%), social educators working in mental health settings (15%), psychiatrists (12%), mental health social workers (7%), occupational therapists (4%), and other allied professionals including primary care doctors and nurses and non-specialists working with mental health service users such as community health workers and administrative staff (15%). The average age was 40.13, ranging from 23 to 65 years of age. Approximately 77% of the sample were women.

Participants in the study were gathered from a pool of professionals who had participated in discussions, training sessions and awareness activities on mental health service user’s rights. The scale was designed to be used as a baseline and follow-up measure.

The study received ethical clearance from the University of Barcelona institutional review board (IRB00003099). All participants gave informed consent and the questionnaires were completed anonymously.

### 2.3. Analysis

Before analysing the data, we carried out a search for outliers by calculating the mean of all the responses for each participant. We excluded a total of seven questionnaires from the analysis as they were indicative of extreme values (most answers corresponding to one of the Likert scale anchor points). 

We calculated frequencies, asymmetry, and kurtosis parameters, as well as item-total correlations for each item in order to decide upon their inclusion in exploratory and confirmatory factor analyses (EFAs and CFAs, respectively), in addition to univariate and multivariate Item Response Theory (IRT) exploratory and confirmatory analyses. EFAs were combined with CFAs through the identification of stable and theoretically congruent dimensions appearing in consecutive principal components analyses, for which fit could be tested through structural equation modelling. In parallel, the discriminative capacity of each item was tested using uni- and multivariate IRT analyses. Through the analysis of item-total correlations, EFA, and CFA factor loading valences, items that indicated that they should be reversed were recoded, with higher scores indicating larger presence of negative attitudes or beliefs (see italicised items in Appendix A). Finally, reliability was tested using Cronbach’s alpha. The psych [44], lavaan [45], ltm [46], and mirt [47] packages for the R software [48] were used to compute all the statistical analyses.

## 3. Results

Frequencies, asymmetry, and kurtosis parameters for each original item can be seen in Appendix A. Due to their low discriminative capacity, we decided to remove items with an asymmetry and kurtosis greater than 1 or less than −1 (eight items) and/or 90% of the cases included in one of the two halves of the Likert scale (nine additional items). We then calculated item-total correlations as well as a unidimensional IRT unconstrained latent variable model with the remaining 27 items. An item on professional pessimism (9), was removed because of nil (*r* = −0.034) correlation with the rest of items and low discrimination parameter (−0.057). All the remaining items had discrimination parameters above 0.5 within a unidimensional IRT model.

Consecutive exploratory factor analyses using Varimax and Oblimin rotations as well as exploratory IRT models were conducted with the 26 remaining items, using the eigenvalue-higher-than-one criterion in CFA and forcing the structure to 2, 3, and 4 factors in CFA and exploratory IRT. This procedure was repeated, temporarily excluding items with low and distributed loadings and low multivariate discriminant parameters. Once we identified a coherent item group (those that tended to remain under the same dimension with high factorial weights and discriminant parameters in different CFA and IRT analyses), we calculated its unidimensionality through Cronbach’s alpha and confirmatory factorial analyses as well as the discrimination parameters of the items. Each group was subsequently removed from the total pool of items and the whole process was repeated with the remaining items, until we obtained a congruent model (19 items) formed by four dimensions.

The dimensions were named as follows: *system criticism*/*justifying beliefs* (items 3, 10, 15, 16, 44), *freedom*/*coercion* (items 4, 6, 23, 34), *empowerment*/*paternalism* (items 1, 8, 27, 28, 38, 40), and *tolerance*/*discrimination* (items 12, 24, 25, 30). The dimensionality of the core model was analysed through confirmatory factor analysis showing a good fit (see Appendix A). Discrimination parameters within a multidimensional IRT (MIRT) were also satisfactory (0.80–2.62). Additional items were added one by one, based on theoretical coherence, factor loadings and discriminant parameters in all EFAs and confirmatory IRT models which were incorporated during the previous process (see Appendix A). We again tested the EFA weights (Table 2), reliability (Table 3), and unidimensionality (Appendix A) for the whole set of 25 items and for each dimension. We also tested the fit of the whole model having added only a specific item (see Table 3 and Appendix A). All items added to each subscale (11, 13, 14, 33, 37, and 39) improved its internal reliability and the fit of the whole model considered as unidimensional, without substantially affecting the fit of the four-dimensional model. As it can be seen in Appendix A, adding item 21 worsened all unidimensionality parameters. Additionally, we considered that it could be included as part of the *tolerance*/*discrimination* subscale. However, it did not improve the reliability of that subscale, nor any of the rest, and hence, it was removed.

Figure 1 shows the CFA path diagram of the final model. Discrimination parameters within the final MIRT were also satisfactory (0.70–2.04, see Table 2). The final scale can be seen in Table 4 (original version in Spanish) and Table 5 (back-translated version in English).

## 4. Discussion

According to our results, the BAMHS may be a useful tool to assess the impact of awareness and training activities on professionals’ beliefs and attitudes towards service users’ rights. This new scale offers flexibility and assumes no prior awareness or knowledge, making it especially suitable for its use in areas where user-led and progressive professional movements are carrying out activities with professionals without previous recovery knowledge or awareness of user rights violations.

The results illustrate four final dimensions, namely: *system criticism*/*justifying beliefs*, *freedom*/*coercion*, *empowerment*/*paternalism* and *tolerance*/*discrimination*. The scale can be scored conveniently in any direction, with higher scores signifying higher respect or a higher violation of rights. We simply advise potential users to make it clear in the methodology of their research report. The final structure of the BAMHS showed an adequate fit according to CFA parameters, good reliability, and good discrimination parameters. Adding six items to the core model did not substantively affect the overall fit of the model, nor that of each of the modified dimensions or the discrimination capacity of each of the items. Additionally, none of the items that were included worsened the reliability of each dimension or the whole model.

The first dimension of the BAMHS materialises the professional beliefs that health-related professionals have which justify the status quo. Claiming that mental disorders are diseases like any other, that their aggressiveness is due to their mental disorders, that it is not possible to recover without the intervention of a professional, and even that some patients will never recover are statements that reinforce the need for mental health staff and their interventions. Regarding the former topic, some authors have stressed the role of biological and genetic attributions in the process of stigmatisation, including the belief that most mental disorders are chronic conditions [54,55]. In some way, understanding that mental disorders are unrecoverable biological conditions might tip the moral balance towards the justification of coercion [56]. Accordingly, professionals scoring high on this subscale might also think that they only use these measures when necessary. In this context, declaring someone incapacitated might be considered an adequate way of care. Finding a justification for the use of extraordinary measures in the very nature of mental disorders might facilitate the concealment between the use of such measures and stating that “mental health service users now have the same rights as other people” and that “mental health professionals, in general, work collaboratively with patients”. These types of assertions are related to the complacency usually found among some mental health professionals despite the continuous use of coercion [57,58,59].

The *freedom*/*coercion* dimension addresses recurrent topics with mental health professionals when discussing service users’ rights. The subscale includes questions on involuntary hospitalization, mechanical restraints, and, inversely, respect for service users’ autonomy. We would like to highlight that more than half of our sample believed that mechanical restraints are sometimes necessary and that one should be involuntarily hospitalised even if they do not pose a threat to others. This is in contrast to the evidence that shows a worse prognosis [60], iatrogenesis [61,62], and even death [63] for people subjected to such coercive measures. Conversely, restraint reduction has been shown to be feasible [64] and to reduce the risk of injury and medical leave among nursing staff [65].

The next subscale, *empowerment*/*paternalism*, represents a series of beliefs related to the supposed inability of people diagnosed with mental disorders to take charge of their lives including having children, making decisions regarding their treatment, or prioritizing treatment over dignity [66,67]. This justifies paternalism in the form of guidelines and constant support, with emotionally distant and value-free practices [68].

Finally, the fourth subscale *tolerance/discrimination* materialises widespread prejudices towards mental health service users. Discrimination occurs in different contexts; for instance, this can include employment discrimination (as many would not feel comfortable with a diagnosed teacher) which is evidenced through low occupational rates [69]. Likewise, social distance reflects the main reason for the stigma that people with mental health problems experience [70,71]. Other discriminatory practices include access to healthcare [72], and those included in legislation, such as the prohibition to vote [73]. 

The main limitation of this validation study is the use of a convenience sample formed by professionals willing to participate in awareness activities. This may have caused biases, such as social desirability, due to the profile of the participants in the activities in which this validation is contextualised. However, this scale is designed to evaluate changes in professionals willing to participate in activities where patients’ rights are discussed. Therefore, we believe that it can be a useful tool to evaluate awareness activities in the mental health field. Future work should culturally and linguistically adapt the tool for other territories and establish psychometric properties.

We believe that the BAMHS, a relatively brief scale tested in diverse mental health provision contexts with a wide range of professionals, can be used to measure the impact of recovery and anti-stigma Targeted, Local, Credible, Continuous Contact (TLC3) methodology-based interventions [74] carried with mental health professionals.

## 5. Conclusions

In conclusion, we believe that our instrument brings a new perspective to the measurement of beliefs and attitudes of mental health professionals in the context of the new era opened by the Convention on the Rights of Persons with Disabilities [75].

## Figures and Tables

**Figure 1 ijerph-16-00244-f001:**
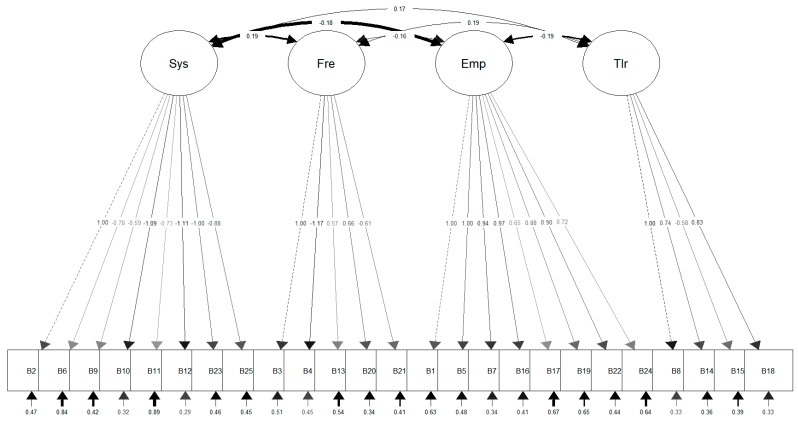
Structural equations diagram of the scale.

**Table 1 ijerph-16-00244-t001:** Previous measures of mental health professionals’ beliefs and attitudes towards service users’ rights.

Measure	Applies to	Constructs Measured	Reference
Custodial Mental Illness Ideology Scale	Mental health professionals	Custodial and humanistic ideologies	[11,12]
Opinions about Mental Illness Scale	Mental health professionals	Public stigma	[13]
Community Attitudes towards the Mentally Ill Scale	General public	Public stigma	[16]
Attitudes towards Psychiatry (ATP-30)	Mental health professionals	Attitudes towards the psychiatry specialty	[30]
Perceived Devaluation and Discrimination Scale	General public/mental health service users	Public stigma	[20]
Affective Reaction Scale	General public	Public stigma	[18,49]
Dangerousness Scale	General public	Public stigma	[18,49]
“Changing Minds” questionnaire	General public	Public stigma	[21]
Recovery Attitudes Questionnaire (RAQ-7)	Mental health professionals	Recovery attitudes	[34]
Professionals’ Beliefs, Goals and Practices in Psychiatric Rehabilitation	Mental health professionals	Recovery practice	[36]
Medical Condition Regard Scale	Medical students	Professional stigma	[27]
Attribution Questionnaire	General public	Public stigma	[15]
Internalized Stigma of Mental Illness (ISMI)	Mental health service users	Public stigma	[23]
Recovery-Oriented Practices Index (ROPI)	Mental health professionals	Recovery practice	[41]
Recovery Self-Assessment	Mental health institutions (professionals, service users and relatives)	Recovery practice	[42,50,51,52]
Recovery Knowledge Inventory (RKI)	Mental health professionals	Recovery knowledge	[31]
Implicit Stigma (Implicit Association Test)	General public Mental health professionals	Public/professional stigma	[22] [25,40]
Integrated Dual Disorders Treatment Model Knowledge Scale	Mental health professionals	Recovery knowledge	[33]
The Project GREAT Recovery Knowledge Measure—Recovery Attitudinal Pre-Post Survey	Mental health professionals	Recovery knowledge	[32]
Mental Health Knowledge Schedule	General public	Public stigma	[19]
Mental Illness: Clinicians’ Attitudes Scale (MICA)	Health professionals	Professional stigma	[26,53]
Police Contact Experience Scale	Police officers	Professional stigma	[29]
Reported and Intended Behaviour Scale (RIBS)	General public		[17]
Quality Indicator for Rehabilitative Care (QuIRC)	Mental health professionals	Recovery practice	[38]
Opening Minds Stigma Scale for Health Care Providers (OMS-HC)	Primary care professionals	Professional stigma	[24]
Consumer Optimism Scale	Mental health professionals	Professional optimism	[33]
Provider Expectations for Recovery Scale	Mental health professionals	Professional optimism for recovery	[35]
PAREM (Attitude Questionnaire developed by Psychiatric Investigations and Education Center)	Mental health students	Professional stigma	[28]
Strengths Model Attitudes Questionnaire (SMAQ)	Mental health professionals	Recovery practice	[37]

**Table 2 ijerph-16-00244-t002:** Factorial weights and discrimination data of the final item pool.

Original Item Number	Final Item Number	Statement	EFA Weights *						
			System criticism/justifying beliefs	Freedom/coercion	Empowerment/paternalism	Tolerance/discrimination	Univariate IRT discriminant parameter within the whole scale	Univariate IRT discriminant parameter within the subscale	Multivariate IRT discriminant parameter
		System criticism/justifying beliefs							
3	2	It is possible to recover without professional interventions.	0.292				1.229	1.091	1.088
10	6	Mental disorders are diseases like any other.	0.257 *				0.639	0.853	0.856
13	9	When patients behave aggressively it is due to their mental disorder.	0.639				0.867	0.845	0.832
14	10	Declaring someone with a severe mental disorder incapacitated is a good way of taking care of that person.	0.471				1.867	1.444	1.422
15	11	Individuals with mental disorders currently have the same rights as other people.	0.463				0.754	0.986	0.991
16	12	Coercive measures are currently applied only when necessary.	0.552				1.929	2.630	2.677
39	23	Some patients will never be able to recover.	0.210 *				1.332	0.955	0.945
44	25	For the most part, mental health professionals work collaboratively with patients.	0.504				1.098	1.465	1.484
		Freedom/coercion							
4	3	People should not be involuntarily hospitalised if they do not pose a threat to the integrity of others		0.638			1.066	1.584	1.644
6	4	Sometimes it is necessary to mechanically restrain patients.		0.557			1.367	1.967	1.907
23	13	When a patient behaves aggressively it is due to the situations, that occur for example in involuntary admissions.		0.757			.577	1.011	1.016
34	20	Greater importance should be placed on promoting the patient’s independence than on reducing the patient’s symptoms.		0.367			1.264	1.166	1.187
37	21	If there are not enough staff, mechanical restraints are the only way to manage violent situations.		0.137 *			1.058	0.898	0.892
		Empowerment/paternalism							
1	1	The possibility of people with severe mental disorders having children should be regulated.			0.496		1.208	1.311	1.318
8	5	Patients with severe mental disorders require clearer instructions than other patients.			0.359		1.254	1.339	1.346
11	7	Professionals should have more say than patients in making treatment decisions.			0.115 *		1.969	1.536	1.585
27	16	People with severe mental disorders always require support to be able to live independently.			0.416		1.408	1.494	1.500
28	17	Objective tests should be prioritised over the professionals’ and patients’ opinion.			0.591		0.632	0.698	0.699
33	19	Respecting the patients’ dignity is important, but some aspects of treatment may require flexibility.			0.240		1.022	1.016	1.032
38	22	When dealing with patients it is important for me not to get emotionally involved.			0.617		1.307	1.482	1.472
40	24	In my clinical practice I try to leave my personal values aside.			0.538		0.812	0.877	0.888
		Tolerance/discrimination							
12	8	Individuals incapacitated by severe mental health problems should have the right to vote.				0.474	1.845	1.541	1.476
24	14	I would feel comfortable making friends with someone with a severe mental disorder.				0.534	1.148	1.786	1.827
25	15	I am uncomfortable with patients who regularly use emergency services.				0.634	0.850	1.121	1.150
30	18	I would be comfortable if a person with a mental disorder were a teacher in a school.				0.602	1.443	1.954	2.035

* Items marked with an asterisk were the only four that weighted higher within a different dimension when performing exploratory factor analysis (EFA) with the final item set. IRT: Item Response Theory; EFA: exploratory factor analysis.

**Table 3 ijerph-16-00244-t003:** Reliability of the core structure, total scale, and four final dimensions including additional items *.

Dimensions	Cronbach’s Alpha
Core structure (19 items)	0.824
Total structure (25 items)	0.867
Dimension 1—System criticism/justifying beliefs	
Core (5 items: 3/2 It is possible to recover without professional interventions, 10/6 Mental disorders are diseases like any other, 15/11 Individuals with mental disorders currently have the same rights as other people, 16/12 Coercive measures are currently applied only when necessary, 44/25 For the most part, mental health professionals work collaboratively with patients)	0.644
Core + 13/9 (When patients behave aggressively it is due to their mental disorder).	0.644
Core + 14/10 (Declaring someone with a severe mental disorder incapacitated is a good way of taking care of that person).	0.681
Core + 39/23 (Some patients will never be able to recover).	0.649
Final (8 items)	0.706
Dimension 2—Freedom/coercion	
Core (4 items: 4/3 People should not be involuntarily hospitalised if they do not pose a threat to the integrity of others, 6/4 Sometimes it is necessary to mechanically restrain patients, 23/13 When a patient behaves aggressively it is due to the situations that occur, for example in involuntary admissions, 34/20 Greater importance should be placed on promoting the patient’s independence than on reducing the patient’s symptoms).	0.641
Core + 37/21 (If there are not enough staff, mechanical restraints are the only way to manage violent situations).	0.652
Dimension 3—Empowerment/paternalism	
Core (6 items: 1/1 The possibility of people with severe mental disorders having children should be regulated, 8/5 Patients with severe mental disorders require clearer instructions than other patients, 27/16 People with severe mental disorders always require support to be able to live independently, 28/17 Objective tests should be prioritised over the professionals’ and patients’ opinion, 38/22 When dealing with patients it is important for me not to get emotionally involved, 40/24 In my clinical practice I try to leave my personal values aside).	0.658
Core + 11/7 (Professionals should have more say than patients in making treatment decisions).	0.689
Core + 33/19 (Respecting the patients’ dignity is important, but some aspects of treatment may require flexibility).	0.676
Final (8 items)	0.709
Dimension 4—Tolerance/discrimination	
Core/Final (4 items: 12/8 Individuals incapacitated by severe mental health problems should have the right to vote, 24/14 I would feel comfortable making friends with someone with a severe mental disorder, 25/15 I am uncomfortable with patients who regularly use emergency services, 30/18 I would be comfortable if a person with a mental disorder were a teacher in a school).	0.650

To facilitate interpretation, both item numbers appear (original/final).

**Table 4 ijerph-16-00244-t004:** Spanish Version of the Beliefs and Attitudes towards Mental Health Service Users’ Rights Scale including scoring details.

*En relación a pacientes con problemas de salud mental y como profesional pienso que*…	1 Totalmente en desacuerdo	2 En desacuerdo	3 De acuerdo	4 Totalmente de acuerdo
1. Se debería regular la posibilidad de que las personas con trastornos mentales severos tengan hijos.	□	□	□	□
2. Es posible recuperarse sin la intervención de un profesional.	□	□	□	□
3. No se debería hospitalizar involuntariamente si no hay peligro para la integridad de terceras personas.	□	□	□	□
4. Las contenciones mecánicas a veces son necesarias.	□	□	□	□
5. Los pacientes con trastorno mental severo necesitan pautas más claras que el resto.	□	□	□	□
6. Los trastornos mentales son enfermedades como cualquier otra.	□	□	□	□
7. Los profesionales debemos tener mayor prioridad que los pacientes para tomar decisiones sobre su tratamiento.	□	□	□	□
8. Las personas incapacitadas por problemas de salud mental severos deberían tener derecho a votar.	□	□	□	□
9. Cuando un paciente lleva a cabo una agresión es por causa de su trastorno mental.	□	□	□	□
10. Una incapacitación es una buena manera de hacerse cargo de una persona con un trastorno mental severo.	□	□	□	□
11. Las personas con trastorno mental tienen hoy en día los mismos derechos que el resto de personas.	□	□	□	□
12. Actualmente se aplican medidas coercitivas sólo cuando es necesario.	□	□	□	□
13. Cuando un paciente lleva a cabo una agresión es por causa de las situaciones que se dan por ejemplo en los ingresos involuntarios.	□	□	□	□
14. Me sentiría cómodo haciéndome amigo de alguien con un trastorno mental severo.	□	□	□	□
15. Me incomodan los pacientes que frecuentan los servicios de urgencias.	□	□	□	□
16. Las personas con trastorno mental severo siempre necesitan apoyo para poder hacer vida autónoma.	□	□	□	□
17. Deberían priorizarse pruebas objetivas sobre las opiniones de profesional y paciente en el planteamiento de un caso.	□	□	□	□
18. Me sentiría cómodo si una persona con trastorno mental fuera profesor en una escuela.	□	□	□	□
19. El respeto a la dignidad de los pacientes es importante, pero a veces el tratamiento exige ser flexible con algunos aspectos.	□	□	□	□
20. Se debería priorizar la autonomía de los pacientes por encima de la disminución de síntomas.	□	□	□	□
21. Si no hay personal suficiente, las contenciones mecánicas son la única manera de poder gestionar situaciones violentas.	□	□	□	□
22. En el trato con los pacientes para mi es importante no implicarme emocionalmente.	□	□	□	□
23. Algunos pacientes no podrán recuperarse nunca.	□	□	□	□
24. En mi práctica clínica intento dejar mis valores personales de lado.	□	□	□	□
25. La mayor parte de los profesionales de salud mental trabajamos de un modo colaborativo con los pacientes.	□	□	□	□
Puntuación (ítems en cada subescala y valencia): Crítica al sistema/creencias justificativas(+2, −6, −9, −10, −11, −12, −23, −25), Libertad/coerción (+3, −4, +13, +20, −21), Empoderamiento/paternalismo (−1, −5, −7, −16, −17, −19, −22, −24), Tolerancia/discriminación (+8, +14, −15, +18). Antes de realizar cualquier cálculo los ítems negativos deben ser recodificados si se desean obtener puntuaciones que indiquen respeto por los derechos (y viceversa para vulneración). Recomendamos promediar los ítems en cada subescala para que todas las puntuaciones tengan un rango de 1–4.				

**Table 5 ijerph-16-00244-t005:** English version of the Beliefs and Attitudes towards Mental Health Service Users’ Rights Scale including scoring details.

*In relation to patients with mental health problems and as a professional I think that*...	1 Totally disagree	2 disagree	3 Agree	4 Totally agree
1. The possibility of people with severe mental disorders having children should be regulated.	□	□	□	□
2. It is possible to recover without professional interventions.	□	□	□	□
3. People should not be involuntarily hospitalised if they do not pose a threat to the integrity of others	□	□	□	□
4. Sometimes it is necessary to mechanically restrain patients.	□	□	□	□
5. Patients with severe mental disorders require clearer instructions than other patients.	□	□	□	□
6. Mental disorders are diseases like any other.	□	□	□	□
7. Professionals should have more say than patients in making treatment decisions.	□	□	□	□
8. Individuals incapacitated by severe mental health problems should have the right to vote.	□	□	□	□
9. When patients behave aggressively it is due to their mental disorder.	□	□	□	□
10. Declaring someone with a severe mental disorder incapacitated is a good way of taking care of that person.	□	□	□	□
11. Individuals with mental disorders currently have the same rights as other people.	□	□	□	□
12. Coercive measures are currently applied only when necessary.	□	□	□	□
13. When a patient behaves aggressively it is due to the situations that occur, for example in involuntary admissions.	□	□	□	□
14. I would feel comfortable making friends with someone with a severe mental disorder.	□	□	□	□
15. I am uncomfortable with patients who regularly use emergency services.	□	□	□	□
16. People with severe mental disorders always require support to be able to live independently.	□	□	□	□
17. Objective tests should be prioritised over the professionals’ and patients’ opinion.	□	□	□	□
18. I would be comfortable if a person with a mental disorder were a teacher in a school.	□	□	□	□
19. Respecting the patients’ dignity is important, but some aspects of treatment may require flexibility.	□	□	□	□
20. Greater importance should be placed on promoting the patient’s independence than on reducing the patient’s symptoms.	□	□	□	□
21. If there are not enough staff, mechanical restraints are the only way to manage violent situations.	□	□	□	□
22. When dealing with patients it is important for me not to get emotionally involved.	□	□	□	□
23. Some patients will never be able to recover.	□	□	□	□
24. In my clinical practice I try to leave my personal values aside.	□	□	□	□
25. For the most part, mental health professionals work collaboratively with patients.	□	□	□	□
Scoring (items in each subscale and valence): System criticism/justifying beliefs (+2, −6, −9, −10, −11, −12, −23, −25), Freedom/coercion (+3, −4, +13, +20, −21), Empowerment/paternalism (−1, −5, −7, −16, −17, −19, −22, −24), Tolerance/discrimination (+8, +14, −15, +18). Before performing any calculations, negative items must be recoded if scores indicating respect for rights want to be obtained (and vice versa for violation). We recommend averaging the items in each subscale so that all scores have a range of 1–4.				

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
