# Peer review of "An Instrument to Measure Mental Health Professionals’ Beliefs and Attitudes towards Service Users’ Rights"

_ijerph, 2019, doi:10.3390/ijerph16020244_

Round 1
Reviewer 1 Report
The article addresses the important topic of professionals’ attitudes towards mental health service users by developing an instrument to measure their attitudes. I suggest modify reference to its appropriateness as a longitudinal measure (eg p3), since it was developed only with point in time data, so its sensitivity to change has not been tested. It would have been preferable to frame the topics, particularly the four dimensions in the positive, rather than currently in the negative, so as not to further stigmatise people. The article presents all the results of all stages with very big tables and long reference list, which is a bit overwhelming, and does not leave room for the final instrument, which would be better as the priority. The language over mental health consumers/users/patients/people/they/one/individuals is inconsistent (eg Table 2) and sometimes disrespectful. I suggest make consistent to an international norm and use that in the final instrument. Table 2 note mentions italics which don’t appear in the table.
I cannot comment on the rigour of the quantitative analysis.
It should be edited for academic English.
Author Response
Dear reviewer,
We would like to resubmit the manuscript entitled “An instrument to measure mental health professionals’ beliefs and attitudes towards service users’ rights”. We are very thankful for all your insightful comments. We have tried to address all the issues raised and have accommodated and incorporated all necessary revisions. Below you can find all points raised and our respective annotation to the modification. We will upload first a clean version of the manuscript, tables and figures, followed by a tracked changes version of the modified documents in the editorial application. We look forward to your comments.
With best regards
Francisco José Eiroa-Orosa
Review table
Reviewer | Our Response |
The article addresses the important topic of professionals’ attitudes towards mental health service users by developing an instrument to measure their attitudes. | Thank you very much |
I suggest modify reference to its appropriateness as a longitudinal measure (eg p3), since it was developed only with point in time data, so its sensitivity to change has not been tested. | Thank you for the observation. We have deleted references to longitudinal information. |
It would have been preferable to frame the topics, particularly the four dimensions in the positive, rather than currently in the negative, so as not to further stigmatise people. | Thank you. We have done so, giving the option to the questionnaire users to conceptualise in one way or the other. |
The article presents all the results of all stages with very big tables and long reference list, which is a bit overwhelming, and does not leave room for the final instrument, which would be better as the priority. | Thank you for this great suggestion. The final instrument, with different parameters is now shown in table 2. The old table 2 has been moved to the annexes. |
The language over mental health consumers/users/patients/people/they/one/individuals is inconsistent (eg Table 2) and sometimes disrespectful. I suggest make consistent to an international norm and use that in the final instrument. | We have done our best to do this consistent in the manuscript. We cannot change the word ‘patient’ from the items as they have been already tested and they are done using professionals’ language (i.e. we need professionals to think about these phenomena the way they usually do). |
Table 2 note mentions italics which don’t appear in the table.
| Thank you for pointing out that. Table 2 has been moved to the annexes as per your above comment. Now italics appear correctly. |
I cannot comment on the rigour of the quantitative analysis. | The methodology has been reviewed by an expert statistician. |
It should be edited for academic English.
| The article has been reviewed by a native speaker. |
Reviewer 2 Report
Important topic. I appreciate the authors' drawing attention to professionals' attitudes to the rights of patients/persons served as a challenge that needs to be addressed, and pointing out the lack of a validated and accepted tool for measuring those attitudes. Such a tool is needed to assess and fix this problem. The paper as written, however, needs some work. 1) It is over-written - too long and provides more than the reader needs - not just in the 76 references!!! but in the text itself. The section offering criticism of contemporary practice could be much more concise. 2) Editing is needed, especially for the English, in which there are both some errors (I won't try to list them here) and some sentences that are over-wordy, need to be simplified to be more effective. 3) The particular methodology to be used in validating a study like this - the refinement and validation of a scale like this - is beyond my competence and deserves someone specialized in that area.
Author Response
Dear reviewer,
We would like to resubmit the manuscript entitled “An instrument to measure mental health professionals’ beliefs and attitudes towards service users’ rights”. We are very thankful for all your insightful comments. We have tried to address all the issues raised and have accommodated and incorporated all necessary revisions. Below you can find all points raised and our respective annotation to the modification. We will upload first a clean version of the manuscript, tables and figures, followed by a tracked changes version of the modified documents in the editorial application. We look forward to your comments.
With best regards
Francisco José Eiroa-Orosa
Review table
Reviewer | Our Response |
Important topic. I appreciate the authors' drawing attention to professionals' attitudes to the rights of patients/persons served as a challenge that needs to be addressed, and pointing out the lack of a validated and accepted tool for measuring those attitudes. Such a tool is needed to assess and fix this problem. The paper as written, however, needs some work. | Thank you for your review. |
1) It is over-written - too long and provides more than the reader needs - not just in the 76 references!!! but in the text itself. The section offering criticism of contemporary practice could be much more concise. | We have shortened the introduction ostensibly. Now we have 74 references, 38 of them are from past measures (table 1). We hope that 36 references are an acceptable number for a validation paper. |
2) Editing is needed, especially for the English, in which there are both some errors (I won't try to list them here) and some sentences that are over-wordy, need to be simplified to be more effective. | The article has been reviewed by a native speaker. |
3) The particular methodology to be used in validating a study like this - the refinement and validation of a scale like this - is beyond my competence and deserves someone specialized in that area. | The methodology has been reviewed by an expert statistician. |
Reviewer 3 Report
Well written paper, structured coherent and balanced in its theoretical and methodological part. The topic dealt in it is innovative . It raises a key issue such as beliefs of mental health professionals towards users rights and defines and implements a model.
Author Response
Dear reviewer,
We would like to resubmit the manuscript entitled “An instrument to measure mental health professionals’ beliefs and attitudes towards service users’ rights”. We are very thankful for all your insightful comments. We believe that, according to your letter, we do not have to make any changes on your part. However, other reviewers have suggested changes. Thus, we will upload first a clean version of the manuscript, tables and figures, followed by a tracked changes version of the modified documents in the editorial application. We look forward to your comments.
With best regards